# Spatiotemporal imaging of valence electron motion

M. Kübel [1,2], Z. Dube[1], A.Yu. Naumov[1], D.M. Villeneuve [1], P.B. Corkum[1] & A. Staudte[1]

Electron motion on the (sub-)femtosecond time scale constitutes the fastest response in many natural phenomena such as light-induced phase transitions and chemical reactions. Whereas static electron densities in single molecules can be imaged in real space using scanning tunnelling and atomic force microscopy, probing real-time electron motion inside molecules requires ultrafast laser pulses. Here, we demonstrate an all-optical approach to imaging an ultrafast valence electron wave packet in real time with a time-resolution of a few femtoseconds. We employ a pump-probe-deflect scheme that allows us to prepare an ultrafast wave packet via strong-field ionization and directly image the resulting charge oscillations in the residual ion. This approach extends and overcomes limitations in laser-induced orbital imaging and may enable the real-time imaging of electron dynamics following photoionization such as charge migration and charge transfer processes.

[1] Joint Attosecond Laboratory, National Research Council and University of Ottawa, 100 Sussex Drive, Ottawa, ON K1A 0R6, Canada. [2] Department of Physics, Ludwig-Maximilians-Universität Munich, Am Coulombwall 1, D-85748 Garching, Germany. Correspondence and requests for materials should be addressed to M.Küb. (email: matthias.kuebel@uni-jena.de) or to A.S. (email: andre.staudte@nrc-cnrc.gc.ca)

maging electronic dynamics in molecules immediately following photoexcitation is of utmost interest to photochemistry as the first few femtoseconds can determine the fate of ensuing reactions[1–3]. Electronic and nuclear dynamics have been probed with attosecond precision[4,5] by means of high harmonic emission[6–8], laser-induced electron diffraction[2,9,10], and photoelectron holography[11,12]. The aforementioned techniques rely on the recollision mechanism[13,14], where the photoionized electron is driven back to the parent ion by the intense laser field and probes the transient molecular or atomic structure. Recollision-free schemes, such as attosecond transient absorption[15] and sequential double ionization have also been used to follow electronic[16–18] and nuclear dynamics[19] on a few-femtosecond time scale.

Attosecond technology not only offers unprecedented time-resolution for ultrafast processes, but also laser-based approaches to imaging electronic structure. Such images can be obtained indirectly by analyzing the high harmonic spectrum as a function of molecular alignment with respect to the laser polarization[7,20,21], or directly, by measuring the photoelectron angular distribution in the molecular frame[9,22–24]. Photoelectron angular distributions have been studied to follow electron dynamics on a sub-picosecond time scale[25,26]. However, the direct imaging of bound electron wave packets on the femtosecond time-scale has yet to be accomplished.

Some of the simplest bound electron wave packets that can be prepared by strong-field ionization are spin–orbit wave packets in noble gas ions[15,27–29]. As the spin–orbit wave packet evolves, the $3p^{-1}$ electron-hole in the noble gas ion oscillates between the $m = 0$ state and the $|m| = 1$ states ($m$ being the magnetic quantum number). This oscillation leads to a time-dependent modulation in the angle-dependent tunnel ionization probability of the ion[16]. Time-resolved measurements of the momentum distribution of photoelectrons, emitted from the ion, would allow for directly imaging the evolving electron-hole[28]. The main obstacle is the contamination of the signal with photoelectrons from the pump pulse[17].

Here, we demonstrate the direct imaging of electron density variations with a temporal resolution of only a few femtoseconds. We prepare a bound wave packet in an argon ion using optical tunnel ionization by a few-cycle visible laser pulse. The resulting multi-electron wavepacket is then imaged via another tunnel ionization process induced by a second few-cycle visible laser pulse. Contamination of the probe pulse signal is avoided by superimposing a weak, orthogonally polarized, carrier-envelope phase-stable, mid-IR streaking field[30] onto the probe pulse. This allows us to separate the primary and secondary photoelectrons spatially and thereby enables direct imaging of the valence-shell wave packet. By inverting the resulting 2D momentum spectra we obtain the autocorrelation functions of the spatial density of the bound electron wavepacket, as seen through the optical tunnel.

## Results

**Time-resolved orbital imaging experiment.** Figure 1 shows a schematic of our pump-probe experiment. In the pump step, strong field ionization of neutral Ar with a few-cycle visible laser pulse causes the coherent population of the $^2P_{3/2}$ and $^2P_{1/2}$

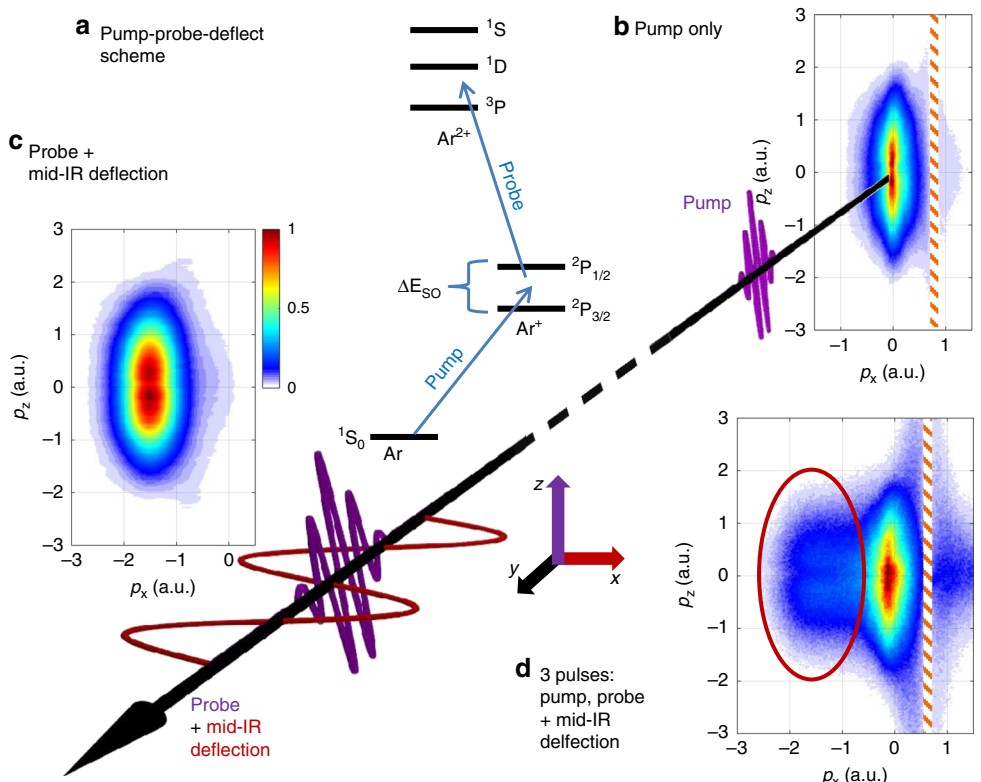

**Fig. 1** Schematic of the time-resolved orbital imaging experiment. **a** A coherent electron wave packet is prepared in Ar$^+$ via strong-field ionization by a few-cycle pump pulse. As the wave packet evolves, a hole (vacancy) oscillates between the $m = 0$ and $|m| = 1$ states of the valence shell of the Ar$^+$ ion with the spin orbit period $T_{SO} = 23.3$ fs[16]. The electron density in Ar$^+$ is probed after a variable time delay using a few-cycle probe pulse in the presence of a phase-stable, mid-infrared deflection field. The deflection field makes the centered momentum distributions produced by the pump pulse alone (**b**) distinguishable from the off-center distribution produced by the probe pulse (**c**). Panel **c** was generated by simulating the effect of the deflection field on the data presented in **b**. In the total electron distribution shown in **d**, the signal with $p_x < -0.5$ a.u. is dominated by electrons from the probe pulse, as indicated by the red oval. The colorscale indicates the electron yield. Each panel is normalized to its maximum. Source data are provided as a Source Data file

fine-structure states. The resulting spin–orbit wavepacket oscillates with a period $T_{SO} = h/\Delta E_{SO} = 23.3$ fs and is probed at a variable time delay using strong field ionization by a second few-cycle visible laser pulse. Superimposed on the probe pulse is an orthogonally polarized, mid-infrared (mid-IR), 40 fs pulse, that deflects and thereby labels the electron created by the probe pulse.

Three-dimensional ion and electron momenta are measured in coincidence using Cold Target Recoil Ion Momentum Spectroscopy (COLTRIMS). We make use of the fact that the few cycle pulse alone produces photoelectrons with a narrow momentum distribution along $p_z$ centered at zero momentum (Fig. 1b). When the orthogonally polarized mid-IR deflection field is superimposed, the ionized electron wavepacket is shifted, as shown in Fig. 1c. In the experiment with all three pulses (Fig. 1d), the probe pulse signal dominates for negative momenta along the direction of the deflection field, as marked by the red oval. In the following, we present results for electrons selected accordingly, see Methods for details.

**Snapshots of an electronic wave packet**. Figure 2 shows our experimental results. Figure 2a shows the delay dependent $Ar^{2+}$ yield for one oscillation of the valence shell wave packet. Measured data for several oscillations are shown in Supplementary Figure 1. We observe a strong modulation of the $Ar^{2+}$ yield of $\approx 45\%$. Ionization is favored when the $m = 0$ state is populated with two electrons. In this case, ionization from $|m| = 1$ state is

negligible[28]. On the other hand, at the yield minima, ionization from the donut shaped $|m| = 1$ orbital becomes significant.

In Fig. 2b and Supplementary Movie 1, we present a time series of measured electron density plots for the wave packet in $Ar^+$. Each density plot is a normalized difference between the delay-dependent and delay-averaged momentum distributions. The series of snapshots shows a narrow spot in the center of the distributions for $\Delta t = 1/2T_{SO}$, corresponding to a yield maximum. The central spot becomes weaker for larger delays, $\Delta t = 4/6T_{SO}$. Eventually, the center spot disappears and a ring around the origin is established at $\Delta t = 5/6T_{SO}$, which appears with maximum brightness at $\Delta t = T_{SO}$, at the minimum of the $Ar^{2+}$ yield.

The experimental images agree qualitatively with the simple calculations at $\Delta t = 1/2T_{SO}$ and $\Delta t = T_{SO}$, respectively. At intermediate values $\Delta t = 4/6T_{SO}$, and $\Delta t = 5/6T_{SO}$, the images are essentially identical to the ones at $\Delta t = 1/2T_{SO}$, and $\Delta t = T_{SO}$, respectively, but exhibit a reduced contrast, see Supplementary Figure 2. For the calculated momentum distributions, we use spatial $Ar^+$ valence orbitals for $m = 0$ and $|m| = 1$, and calculate the transversal momentum space orbitals by Fourier transform, see Supplementary Method 2 for details. Plotted in Fig. 2 are the normalized differences between the $m = 0$ and $|m| = 1$ vacancy states.

The expected circular symmetry of the momentum distributions is broken by a noticeable stretch along the $p_x$ axis. This distortion arises from the mid-IR deflection field, which is used

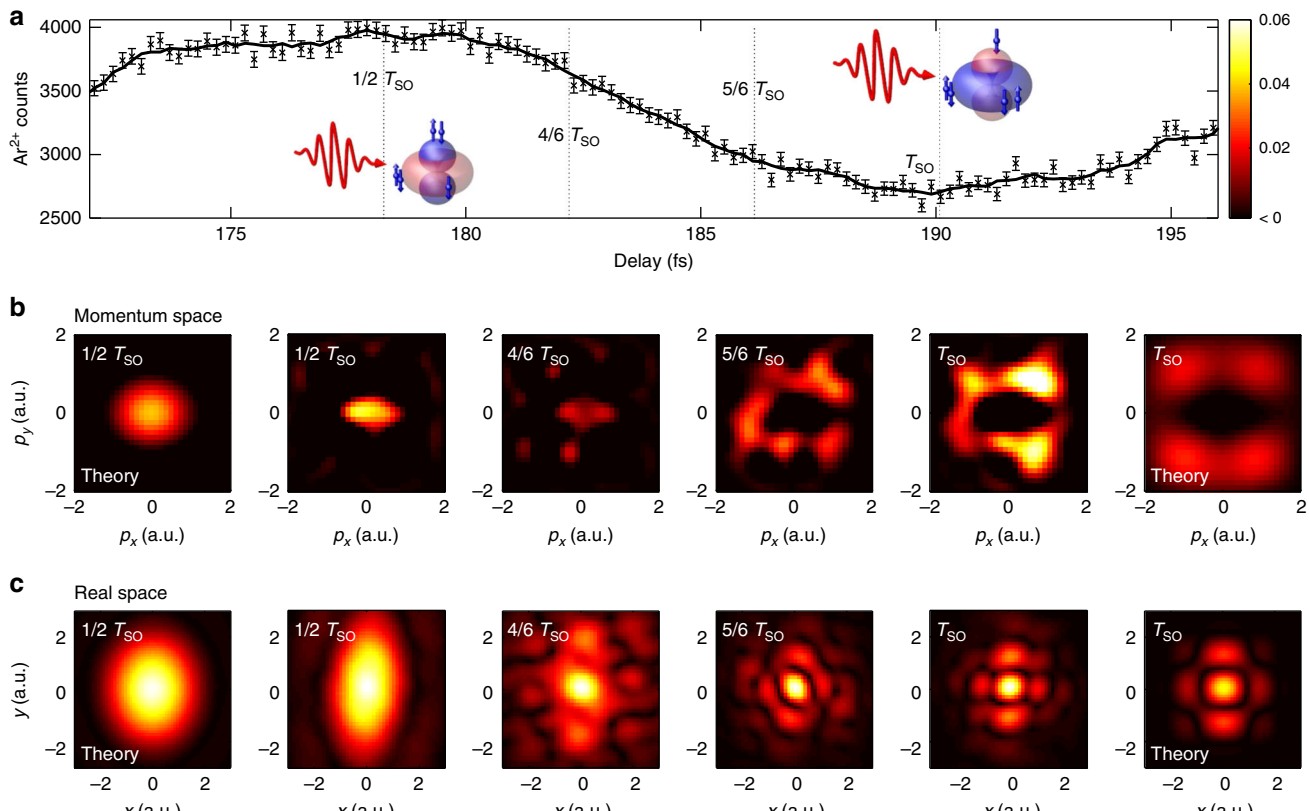

**Fig. 2** Snapshots of a spin–orbit wave packet in the argon cation. **a** Measured $Ar^+$ yield as a function of time delay between pump and probe pulses. The cartoons illustrate the electron configuration in $Ar^+$ at the time of interaction with the probe pulse. For the positions marked with dotted lines and fractions of the spin–orbit period, $T_{SO}$, we present measured electron density plots in momentum space (**b**) and real space (**c**). The momentum space images show the positive part of the normalized differences of delay-dependent and delay-averaged electron momentum distributions. The data has been integrated over $p_z$ and a delay range of ±3 fs. A low pass frequency filter has been applied. The real space images show the Fourier transform of the momentum space images. The theory plots show the normalized differences between calculated $Ar^+$ momentum space orbitals corresponding to $m = 0$ and $|m| = 1$ vacancies, and their Fourier transforms, respectively. Source data are provided as a Source Data file

to identify the probe pulse signal. The stretch in momentum space corresponds to a contraction in the real space images. Because the $x$ and $y$ directions are equivalent, the distortion does not cause loss of information. The mean momentum shift induced by the deflection field, $\Delta p_x = -1.5$ a.u., has been subtracted from the presented images.

In Fig. 2c, we show the autocorrelation functions of the spatial electron density, which are obtained from the momentum distributions by Fourier transform, assuming a flat phase. The spatial distributions obtained from the experimental data qualitatively agree with the theoretical results. This indicates that our methods allows for reconstruction of real-space features of the time-dependent valence electron density.

**Longitudinal momentum distribution.** Next, we turn our attention to the photoelectron momentum component along the ionizing laser field, the z-direction. In Fig. 3, we examine the distributions in the $(p_y/p_z)$ plane.

The spectra recorded at maximum $(\Delta t = 1/2 T_{SO})$ and minimum $(\Delta t = T_{SO})$ $Ar^{2+}$ yields are qualitatively indistinguishable, see Fig. 3a. The normalized difference of the two spectra reveals the distinctions between the momentum distributions arising from ionization of $m = 0$ and $|m| = 1$ states and is presented in Fig. 3b. A clear pattern is visible: the blue areas at larger perpendicular momenta $(|p_y| \gtrsim 0.5 \, a.u.)$ indicate the contribution of the donut shaped $|m| = 1$ orbital at the yield minima, as seen at $\Delta t = T_{SO}$ in Fig. 2b. The red area at small perpendicular momenta $(|p_y| < 0.5 \, a.u.)$ indicates the dominance of ionization from the $m = 0$ orbital at the yield maxima, as seen at $\Delta t = 1/2 T_{SO}$ in Fig. 2b.

Strikingly, the normalized difference exhibits pronounced maxima for large longitudinal momenta $(|p_z| > 2 \, a.u.)$. Similar observations have been made in pump-probe experiments on double photodetachment from negative ions[31,32]. The maxima observed at large longitudinal momenta raise the question whether the final momentum distributions are, in fact, influenced by the momentum distribution in the bound state. Even though it is intriguing to speculate whether orbital imaging is not purely two-dimensional, as in very recent work on alignment-dependent molecular ionization[33], we offer a different interpretation in Fig. 3c. The plot shows the normalized difference of the longitudinal momentum distributions recorded at the yield

maxima and yield minima. The experimental results are selected for small perpendicular momenta (indicated by the dotted box in Fig. 3b) and compared to the results of a computational model, similar to the one proposed in ref. [28], and detailed in the Methods section. In the model, we calculate the instantaneous non-adiabatic tunnel ionization rates of the $|m| = 1$ and $m = 0$ vacancies. The computational results agree very well with the experimental ones. They indicate that the maxima at large longitudinal momenta arise because the ratio of the ionization probabilities for the two vacancy states varies throughout a laser half cycle.

Specifically, large momenta are produced near the zero crossing of the laser electric field within the optical cycle. At these laser phases, the vector potential is close to its maximum and, correspondingly, the electric field is rather weak. As the $m = 0$ vacancy state is harder to ionize than the $|m| = 1$ vacancy, its ionization probability drops faster with decreasing field strength. Hence, ionization near the zero crossing has an increased contribution from the $|m| = 1$ vacancy. In the fashion of a streak camera, the laser vector potential maps the electron emission times to final momenta, leading to the observed maxima in the normalized difference at large longitudinal momenta.

## Discussion
So far, we have shown that our pump-probe scheme allows us to identify double ionization events where the first and second ionization occurs in the pump and probe pulse, respectively. For these events we can separate the first from the second photoelectron, exploiting the deflection induced by the mid-IR streaking field[30]. Recording the transverse momentum distribution of the second photoelectron enables us to image the electron dynamics unfolding in the cation. We have also shown that the longitudinal momentum distribution of the second photoelectron carries information on the ionization dynamics of the cation in a non-stationary state. In the following, we address the question how quantitative information can be extracted from the measured orbital images.

Figure 4a and Supplementary Movie 2 show the normalized differences of the projected 2D momentum distributions measured at maximum and minimum $Ar^{2+}$ yield. Figure 4b presents calculated distributions, based on a simple imaging model. The

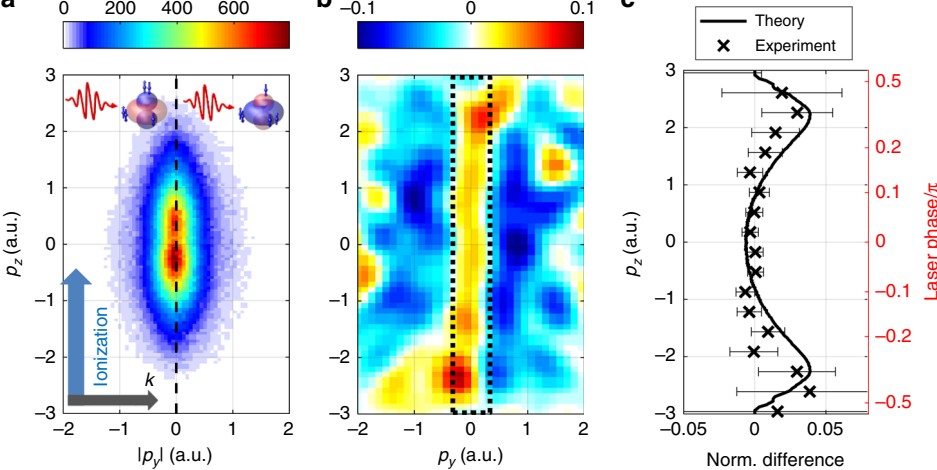

**Fig. 3** Streaking ionization of an electron wave packet in $Ar^+$. **a** Momentum distributions in the $p_y/p_z$ plane for ionization from a coherent wave packet in $Ar^+$. The left (right) half corresponds to delay values with a maximum (minimum) in the $Ar^{2+}$ yield. Each spectrum is normalized to the same number of counts. The normalized difference between the left and right side is displayed in **b**. The dotted box indicates the momentum range for which the normalized difference is plotted along $p_z$ in **c**. The experimental data are compared to the calculated difference in the instantaneous ionization probabilities for the $m = 1$ and $|m| = 0$ vacancies. Error bars are s.d. Source data are provided as a Source Data file

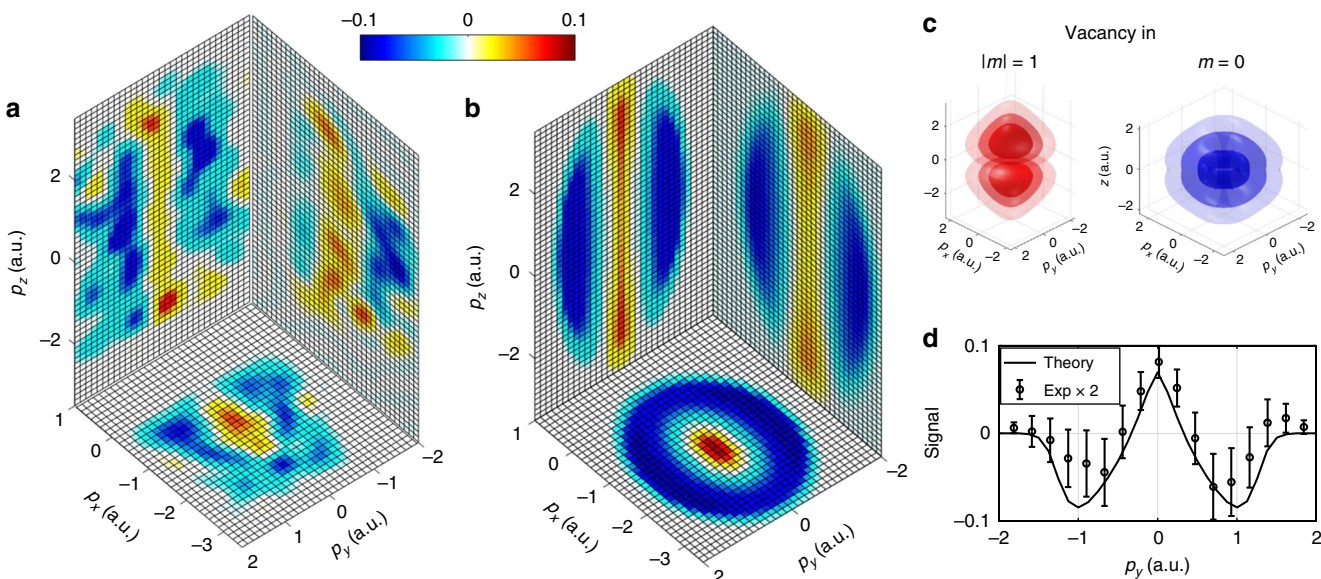

**Fig. 4** Imaging time-dependent valence electron densities. **a** Normalized differences between the momentum distributions recorded for ionization of Ar$^+$ at delays corresponding to yield maxima and yield minima. The data in each plane is integrated over the third dimension. Low-pass frequency filtering has been applied to the experimental data. **b** Results of an analytical imaging model. The color bar applies to both, experimental and theoretical results. **c** Modulus square of the Ar$^+$ 3p orbital wave functions used in the simulation. Blue and red colors are chosen for presentational purposes. **d** Cut through the center of the measured and calculated 3D normalized differences along $p_y$. The experimental signal is multiplied by a factor of 2 to facilitate direct comparison of the measured and calculated widths of the signals. Source data are provided as a Source Data file

model is described in the Methods section. It uses the orbitals depicted in Fig. 4c, i.e., the Ar$^+$ 3p orbitals with $|m| = 1$, and $m = 0$. These electronic density functions are multiplied with a transversal filter function, which describes the tunneling probability as a function of the perpendicular momentum[34]. The acceleration in the combined laser field is simulated, and the sub-cycle streaking effect, discussed above, is taken into account. The simulated momentum distributions exhibit reasonable agreement with the experimental results. In particular, as shown in Fig. 4d, almost quantitative agreement is obtained for the width of the distribution along the laser propagation, which remains unaffected by the laser field after tunneling. This indicates that the model captures the essential imaging mechanism. Thus, valence electron densities in real space could be reconstructed from time-resolved orbital imaging experiments, using appropriate computational techniques. The spatial resolution of the reconstructed real-space distribution is given by the maximum perpendicular photoelectron momentum, which is limited by the transversal filter function and the signal to noise ratio. Time-resolved orbital imaging will greatly benefit from the development of intense few-cycle laser sources approaching the MHz range[35].

Our method offers exciting prospects for time-resolved imaging of molecular orbitals. For example, strong-field ionization can induce purely electronic dynamics, such as charge migration[8], or correlated electron-nuclear dynamics such as dissociation or isomerization. Our pump-probe scheme will enable imaging of the electronic rearrangements that take place during such processes. At the same time, nuclear dynamics and configurations can be tracked with coulomb explosion imaging[36,37] or laser-induced electron diffraction[2,9,10,38].

## Methods

**Experimental setup**. The experiment relies on the set-up developed for sub-cycle tracing of ionization enabled by infrared (STIER), which has been described in ref. [30]. In brief, the output of a 10 kHz, 2 mJ titanium:sapphire laser (Coherent Elite) is split in two parts to obtain 5 fs, few-cycle pulses, centered at 730 nm, from a gas-filled hollow core fiber, and 40 fs, phase-stable mid-IR idler pulses at 2330 nm from an optical parametric amplifier. We extend STIER to pump-probe

experiments by further splitting the few-cycle pulses and recombining them in a Mach-Zehnder interferometer in order to obtain pump and probe pulses with adjustable time delay. The few-cycle pulses pass through a broadband half wave plate and are recombined with the mid-IR pulses. In order to avoid overlap between the mid-IR pulse and the visible pump pulse, the pump-probe delay was offset by $t_0 = 6670$ fs, much larger than the duration of the mid-IR pulses. The offset is not included in the delay values given in the main text. Choosing this large offset is legitimate, as it has been demonstrated that no notable dephasing of the spin–orbit wave packet occurs over the course of several nanoseconds[16]. We have tested in separate experiments that no significant overlap between visible and mid-IR pulse occurs for pump probe delays >50 fs.

The laser pulses (pulse energies of 2.5 μJ for each of the visible pulses, and 18 μJ for the mid-IR pulse) are focused ($f = 75$ mm) into a cold ($T \approx 10$ K) argon gas jet in the center of a COLTRIMS[39]. We estimate the focal spot sizes (1/e$^2$ width) as $7 \pm 2$ μm for the visible pulses and $30 \pm 10$ μm for the mid-IR pulse. Photoelectrons and ions arising from the interaction are detected in coincidence, and their three-dimensional momenta are measured using time and position sensitive detectors. The polarization of the mid-IR deflection field is along the $x$ axis, which is defined by the spectrometer axis of the COLTRIMS. The laser propagates along the $y$ axis and the ionizing few-cycle pulses are polarized along $z$. The electron count rate was kept below 0.2 electron per laser pulse to limit the number of false coincidences. The laser intensity of the visible pulses of $(6.0 \pm 1.0) \times 10^{14}$ W/cm$^2$ was estimated from the carrier-envelope phase-dependent momentum spectra along the laser polarization[40]. The intensity of the mid-IR pulses was estimated from the deflection amplitude $\Delta p_x = -1.5$ a.u. as $3 \times 10^{13}$ W/cm$^2$, low enough to not cause notable ionization of Ar or Ar$^+$.

**Data analysis**. To obtain images of the transient electron density in the Ar$^+$ valence shell, it is crucial to identify the electrons produced in the second ionization step, Ar$^+ \rightarrow$ Ar$^{2+}$+e$^-$ by the probe pulse. This is accomplished as follows. First, recorded electron spectra with and without the deflection field present are compared, see Fig. 1b, d. This shows that for momenta $p_x < -0.5$ a.u., the (deflected) probe pulse signal clearly dominates. We estimate that the contribution of the pump pulse to the signal in the red oval in Fig. 1d as <10% at $p_x = -0.5$ a.u., and ~1% at $p_x = -2$ a.u. Next, we select events for which an Ar$^{2+}$ ion has been detected in coincidence with one electron. The momentum component of the other electron along the deflection field is calculated using momentum conservation. The events in which the second ionization step occurs in the probe pulse are selected with the following conditions,

$$p_x^{meas} < -0.3 \,\text{a.u.,} \tag{1}$$

$$-1 < p_x^{calc} < 0.6 \,\text{a.u.,} \tag{2}$$

where $p_x^{meas}$ is the measured electron momentum component along the IR polarization, and $p_x^{calc}$ is the momentum component calculated from momentum conservation. The rationale for the above conditions is outlined in Supplementary Method 1 and visualized in Supplementary Figure 3.

Having identified the electrons produced in the second ionization step, the electron density plots are obtained by calculating normalized differences of signal, $S$, and reference, $R$, photoelectron momentum distribution:

$$D = (S - R)/(S + R). \qquad (3)$$

**Orbital effect in the longitudinal momentum spectra.** To calculate the ionization rates for the two orbitals we build on the model described in ref. [28]. However, we ignore some ionization pathways with low transition probability. For the simulations, a 5-fs (full width at half maximum of the gaussian intensity envelope) laser pulse with frequency $\omega = 0.06$ a.u. and intensity $I = 1.0 \times 10^{15}$ W/cm$^2$ is used. The ionization probability for either orbital is calculated at every point in time using the rates for non-adiabatic tunneling proposed in ref. [41]. Longitudinal momentum spectra are obtained from the vector potential of the laser pulse, appropriately weighing each contribution with the calculated rate. The calculations are repeated for 16 different values of the carrier-envelope phase (CEP), and the results are averaged over the CEP. The normalized difference of the spectra calculated for $|m| = 1$ and $m = 0$ vacancies is calculated and plotted in Fig. 3c. Usage of ADK formula[42] instead of the non-adiabatic tunneling formula[41], leads to very similar results for the normalized difference of the longitudinal spectra.

**Imaging model.** Here, we give a short description of the procedure used to generate the theoretical images shown in Fig. 4b. A detailed description can be found in Supplementary Method 2. Real space wave functions for the Ar$^+$ valence orbital are taken from the computational chemistry software GAMESS. Momentum-real space wave functions are calculated by partial Fourier transform, as described in ref. [34], and plotted in Fig. 4c.

The wavefunctions squared are multiplied with a "tunnel filter"[34] to obtain the transversal momentum distribution at the tunnel exit. The tunnel filter suppresses large momenta perpendicular to the direction of tunneling. In order to obtain the momentum distributions after propagation in the laser field, the momentum distributions at the tunnel exit are convoluted with Gaussian functions, representing the ionizing visible, and mid-IR deflection fields. The orbital effect in the longitudinal direction is taken into account.

The momentum distribution that correspond to $m = 0$ and $|m| = 1$ vacancies are given by appropriate linear combination of the spectra calculated for the $m = 0$ and $|m| = 1$ wave functions. To calculate the normalized differences in the three momentum planes, the spectra are integrated over the third dimension.

## Data availability

The data for Figs. 1b–d, 2a–c, 3a–c and 4a–d; and Supplementary Figures 1, 2a–c, and 3 are provided as a Source Data file. The data that support the findings of this study are available from the corresponding author upon reasonable request.

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

## Acknowledgements

We thank D. Crane and B. Avery for technical support. We acknowledge fruitful discussions with A. Fleischer, B. Bergues, and M. Spanner. This project has received funding from the EU's Horizon2020 research and innovation program under the Marie Sklodowska-Curie Grant Agreement No. 657544. Financial support from the National Science and Engineering Research Council Discovery Grant No. 419092-2013-RGPIN is gratefully acknowledged.

## Author contributions

M.K. and A.S. conceived and planned the experiment. M.K., Z.D. and A.S. conducted the measurements. M.K., A.Y.N., D.M.V., P.B.C. and A.S. analyzed and interpreted the data. All authors discussed the results and contributed to the final manuscript.

## Additional information

**Competing interests:** The authors declare no competing interests.

