## [Peer Review File · Nature Communications]

Reviewers' comments:

Reviewer #1 (Remarks to the Author):

This manuscript presents a novel approach to experimental time-resolved imaging of electron motion in atoms on femtosecond scale. It combines the method of measuring difference electron momentum spectra (previously used by this group very productively on several occasions) with using an additional long-wavelength pulse to separate the second electron (removed by probe pulse from the ion) spatially from the first (removed by pump pulse from neutral molecule). That is an ingenious (but technically hard to implement) approach which is likely to be adopted by other groups as well. This new method results in a set of snapshot images, which authors correlate with electron-hole time-dependent density evolving with a spin-orbit wave packet in Ar cation. While those wavepackets were produced and detected before by measuring time-dependent ionisation yields, to my knowledge this is the first example of direct imaging of time-dependent electron density. That is a substantial advance which deserves publication in Nature Communications. I have only one concern which I would like to see addressed before I could wholeheartedly recommend this manuscript for publication. My concern is that the presented data cover only a single cycle of the spin-orbit electron oscillation. That leaves open a small possibility that the observed changes could be due not to electron motion, but to some other effect. For instance, it is not clear if the delay between the few-cycle probe pulse and far-IR streaking pulse could also change for any reason. Far-IR pulse is recombined with the visible pulse pair produced by Michelson interferometer and it is not made clear if pump or probe pulse was delayed in respect to the streaking pulse - could be either since range of delays is quite limited vs the large offset between pump and probe pulses. In any case showing just two periods of oscillation with a well known period would completely remove that doubt. I have no other objections - the manuscript is very well and clearly written and presents a significant advance in time-resolved imaging.

Reviewer #2 (Remarks to the Author):

In this manuscript, the authors present a simple but very beautiful and elegant three-pulse pump-probe-deflect measurement approach to (i) pump and photoexcite Argon atoms, (ii) probe the excited atoms by releasing an electron wave packet (EWP), and (iii) deflect the ejected EWP in momentum space such that the authors can tag the signal of interest, avoiding overlap with pump-only and probe-only background signal that typically occurs in two-pulse time-resolved pump-probe measurements. The authors present measured and calculated data to demonstrate that their three-pulse, all-optical approach can extract the Ar+ 3p orbitals in two magnetic quantum states of $m=0$ and $|m|=1$. Their ultrafast three-pulse measurements track the ultrafast dynamics of how an EWP oscillates between these two magnetic states of the 3p atomic orbital in Argon.

My comments on the manuscript are as follows:

- In the abstract, you mention the concept of “probing electron motion inside molecules in real-time”. What is the impact of probing electron motion in real-time? Why is this so important? Can you provide several important examples or references, such as following the initial conditions of a chemical reaction in the first few fs of the reaction where the fate of the reaction is already determined on the electronic (i.e. attosecond) timescale. Maybe you can comment on this either in the abstract or the introduction. It's important to establish why the importance of following electronic motion in real time from the outset of the manuscript.

- In the first paragraph of the introduction, please also include the following reference to “recollision mechanism [18]”: Krause J L, Schafer K J and Kulander K C 1992 High-harmonic generation from atoms and ions in the high intensity regime Physical Review Letters 68 3535.

- In Fig. 1, you do not really explain in detail the significance of the top panel, panel (a) with the atomic orbitals of the two magnetic states. I do not understand how the diagonal dotted lines relate to the pump and MIR deflection pulses as you have schematically shown. What is the point of these dotted diagonal lines? Maybe it is better served placing these orbitals of the two magnetic states into Fig. 2, where you investigate the effect of ionization from $m=0$ and $m=1$ states in more detail? The figure currently is quite overloaded with lots of information, and removing the m state orbital pictures will significantly help. The main message of this figure is that you deflect the momentum distribution when using the third, MIR deflection field, to tag the signal of interest in momentum space.

- Further comment on Fig. 1 - just to clarify: panel (c) shows the electron distribution of probe-only + the MIR field (i.e. 2 pulse signal)? And, if I understand correctly from Fig. 1 and its caption, panel (d) is the 3-pulse signal (pump, probe + MIR deflection fields)? If so, then please change the label of panel (c) from "probe only" to "probe + MIR deflection", and change panel (d) label from "pump+probe" to "3 pulse: pump, probe, MIR deflection" or something similar just to make it clear.

- Fig. 1 data: The effect of the MIR deflection field on the probe-only off-centre electron distribution is remarkably beautiful!

- First sentence in second row of text on Page 2 - please improve the written English as follows: replace “assumes its” with “appears with”.

- Is it not possible to include the breaking of the circular symmetry distortion in the measured momentum maps from the MIR deflection field into your calculations? If so, could you please provide further details and calculated results which incorporate the MIR field into your calculations. These could go into your supplementary online materials (SOM).

- You assign the features in Fig. 3b to $m=0$ and $|m|=1$ orbitals at the yield maxima and minima, respectively. I think it would help the reader a lot if you have a statement clarifying that the normalised difference signal corresponds to the difference between $1/2T_{SO}$ (where $m=0$ dominates) and T_{SO} (where $m=1$ dominates) data, and relating this to the momentum maps shown in Fig. 2b.

- In Fig. 2c, can you comment on why there is a discrepancy between theory and experimental results in the real space data at T_{SO} , particularly along the $y=0$ line. Secondly, at $1/2T_{SO}$, why is your measured distribution compressed along the x-axis?

- At times $4/6$ and $5/6 T_{SO}$ in Fig. 2, do your calculations also reproduce the measured features in both momentum and real space – please could you show these results, at least in the SOM. Can you also comment on the features present in the measured distributions of panels (b) and (c).

- In Fig. 3, you state “selected electrons”, but which selected electrons are you referring to?

- In Fig. 3(c), can you comment on the two outlier measured data points at p_z of 1-2.

- In Fig. 4(d), can you please include error bars in your measured data.

- On page 4, you state “So far, we have shown that we can separate the first from the second photoelectron produced by sequential double ionization.”. Where have you shown this and explicitly discussed this? Is this related to your earlier statement of “This allows us to separate the primary and secondary photoelectrons spatially and thereby enables direct imaging of the valence-shell wave packet.”? Maybe I have missed this, but can you please clarify the statement on sequential double ionization.

- In the next sentence, you state that “the second photoelectron enables us to image the oscillating valence electron density”. However, nowhere in the main text above did you mention that it is the second electron photoelectron momentum distribution data in Fig. 2 and Fig. 3 that shows this. How can you distinguish the signal from the first and second electron in the Ar²⁺ momentum map signals? How can you be so sure and specific that the oscillating valence electron density signal is exclusively from the second photoelectron? It would be helpful for future readers to have this point clarified, at least with a brief explicit sentence.

- The first sentence in the second column of text on page 4: why “partially”? Why only partial reconstruction – can you comment please.

- You mention coulomb explosion imaging (CEI) and laser-induced electron diffraction (LIED) at the end of this paragraph. Can you please include some referenced work, such as:

CEI1 - ultrafast nuclear dynamics: K. Amini et al., Structural dynamics 5, 014301 (2018).

CEI2 - geometric isomer configurations: M. Burt et al., J. Chem. Phys. 148, 091102 (2018).

LIED1 – imaging deprotonation reaction: Wolter2016 reference used in this manuscript.

LIED2 - intra-pulse nuclear-electronic dynamics: K. Amini et al., arxiv:1805.06793.

- In your methods, can you please specify the pulse energy and focal spot size (1/e² width) of your three laser pulses.

Some very minor comments:

- Throughout the manuscript, you use the word “via” quite regularly, particularly in the top-right paragraph on page one. Please avoid so many repetitions of this word in such a short space of text. Also, please make “via” italicized.

- Please make all roman variables in the main text (such as momentum, p) italicized.

Reply to Referee A:

Referee A: *"I have only one concern which I would like to see addressed before I could wholeheartedly recommend this manuscript for publication. My concern is that the presented data cover only a single cycle of the spin-orbit electron oscillation. That leaves open a small possibility that the observed changes could be due not to electron motion, but to some other effect. For instance, it is not clear if the delay between the few-cycle probe pulse and far-IR streaking pulse could also change for any reason. The far-IR pulse is recombined with the visible pulse pair produced by Michelson interferometer and it is not made clear if pump or probe pulse was delayed in respect to the streaking pulse - could be either since the range of delays is quite limited vs the large offset between pump and probe pulses. In any case showing just two periods of oscillation with a well-known period would completely remove that doubt. I have no other objections – the manuscript is very well and clearly written and presents a significant advance in time-resolved imaging."*

In order to provide additional evidence that the yield modulation shown in Figure 2(a) is due to the spin-orbit wave packet, we have added a plot to the supplementary information. The new plot shows the measured delay dependence of the total Ar²⁺ yield.

Supplementary Fig.S1 Measured Ar²⁺ yield as a function of pump-probe delay for several cycles of the spin-orbit period.

In the main text, subsection "Snapshots of an electronic wave packet", we added a reference to Supplementary Fig.1:

"Measured data for several oscillations are shown in Supplementary Fig. 1."

The following text has been added in the SI to describe Figure S1:

"In Supplementary Figure 1 we present the recorded Ar²⁺ yield for all detected coincidences of Ar²⁺ with one electron over the entire delay range studied in the experiment. The modulation period is (23.5 ± 0.2) fs, in agreement with the expected period of 23.3fs. The slightly higher Ar²⁺ yield around Δt = 180 fs is attributed to the small influence of the streaking pulse on the ionization probability. Note that the delay between pump pulse and streaking field is kept constant, while the probe pulse is delayed with respect to the other two pulses. An active stabilization between probe and streaking pulses would allow for delaying the pump pulse instead, avoiding the influence of the streaking pulse envelope on the Ar²⁺ yield."

Reply to Referee B:

Comment 1: *“In the abstract, you mention the concept of “probing electron motion inside molecules in real-time. What is the impact of probing electron motion in real-time? Why is this so important? Can you provide several important examples or references, such as following the initial conditions of a chemical reaction in the first few fs of the reaction where the fate of the reaction is already determined on the electronic (i.e. attosecond) timescale. Maybe you can comment on this either in the abstract or the introduction. It's important to establish why the importance of following electronic motion in real time from the outset of the manuscript.”*

The referee raises a good point, as to why anyone should be interested in observing electron motion. The previous manuscript may have assumed that interest be evident. Therefore we have modified the abstract and introduction to provide additional motivation for our work.

The abstract now begins with the following sentences:

“Electron motion on the (sub-)femtosecond time scale constitutes the fastest response in many natural phenomena such as light-induced phase transitions and chemical reactions. Whereas static electron densities in single molecules can be imaged in real-space using scanning tunnelling and atomic force microscopy, probing real-time electron motion inside molecules requires ultrafast laser pulses. Here, we demonstrate ...”

In the introduction we added another sentence to include a few references for the interested reader. The beginning of the introduction now reads:

“Imaging electronic dynamics in molecules immediately following photoexcitation is of utmost interest to photochemistry as the first few femtoseconds can determine the fate of ensuing reactions [1-3]. Electronic and nuclear dynamics have been probed with attosecond precision [4,5] by means of ...”

Comment 2: *“In the first paragraph of the introduction, please also include the following reference to “recollision mechanism [18]”: Krause J L, Schafer K J and Kulander K C 1992 High-harmonic harmonic generation from atoms and ions in the high intensity regime Physical Review Letters 68 3535..”*

The reference has been included in the revised manuscript.

Comment 3: *“In Fig. 1, you do not really explain in detail the significance of the top panel, panel (a) with the atomic orbitals of the two magnetic states. I do not understand how the diagonal dotted lines relate to the pump and MIR deflection pulses as you have schematically shown. What is the point of these dotted diagonal lines? Maybe it is better served placing these orbitals of the two magnetic states into Fig. 2, where you investigate the effect of ionization from $m=0$ and $m=1$ states in more detail? The figure currently is quite overloaded with lots of information, and removing the m state orbital pictures will significantly help. The main message of this figure is that you deflect the momentum distribution when using the third, MIR deflection field, to tag the signal of interest in momentum space.”*

We agree that figure 1 has been quite crowded. As suggested, we removed the depiction of the $m=0$ and $m=1$ states along with the diagonal lines. Also we rearranged the elements in the figure. We feel that the new version is more accessible than the previous one.

In the caption of figure 1, we removed the sentence

“As the wave packet evolves, a hole (vacancy) oscillates between the $m=0$ and $|m|=1$ states of the valence shell of the Ar^+ ion with the spin orbit period $T_{SO} = 23.3$ fs.”

In the fourth paragraph, we replaced

“As the spin-orbit wave packet evolves, the $3p^{-1}$ electron-hole density in the noble gas ion changes from a peanut shape into a donut shape and back, illustrated in Fig. 1.”

with

“As the spin-orbit wave packet evolves, the $3p^{-1}$ electron-hole in the noble gas ion oscillates between the $m=0$ and $|m|=1$ states (m being the magnetic quantum number).”

Comment 4: *“Fig. 1 data: The effect of the MIR deflection field on the probe-only off-centre electron distribution is remarkably beautiful!”*

We thank the referee for the positive evaluation.

Comment 5: *“First sentence in second row of text on Page 2 - please improve the written English as follows: replace ‘assumes its’ with ‘appears with’.”*

We changed the manuscript as requested.

Comment 6: *“Is it not possible to include the breaking of the circular symmetry distortion in the measured momentum maps from the MIR deflection field into your calculations? If so, could you please provide further details and calculated results which incorporate the MIR field into your calculations? These could go into your supplementary online materials (SOM).”*

We have created a new version of figure 2, where the distortion of the momentum maps caused by the deflection field is included in the calculations. The details on how the calculated momentum maps are obtained are given in the SI, section “Calculated electron density snapshots”. There, we also present results for the intermediate time steps and discuss them (see comment #9).

Comment 7: *“You assign the features in Fig. 3b to $m=0$ and $|m|=1$ orbitals at the yield maxima and minima, respectively. I think it would help the reader a lot if you have a statement clarifying that the normalised difference signal corresponds to the difference between $1/2T_{SO}$ (where $m=0$ dominates) and T_{SO} (where $m=1$ dominates) data, and relating this to the momentum maps shown in Fig. 2b.”*

According to the suggestion, we replaced on page 2

“The spectra recorded at maximum and minimum Ar^{2+} yields are qualitatively indistinguishable, see Fig. 3 (a). However, the normalized difference of the two spectra, presented in Fig. 3(b), reveals a clear pattern: The blue areas at larger perpendicular momenta ($|p_y| > 0.5$ a.u.) indicate the contribution of the donut shaped $|m|=1$ orbital at the yield minima. The red area at small perpendicular momenta ($|p_y| < 0.5$ a.u.) indicates the dominance of ionization from the $m=0$ orbital at the yield maxima.”

with

“The spectra recorded at maximum ($\Delta t = \frac{1}{2} T_{SO}$) and minimum ($\Delta t = T_{SO}$) Ar^{2+} yields are qualitatively indistinguishable, see Fig. 3(a). The normalized difference of the two spectra reveals the distinctions between the momentum distributions arising from ionization of $m=0$ and $|m|=1$ states and is presented in Fig. 3(b). A clear pattern is visible: The blue areas at larger perpendicular momenta ($|p_y| > 0.5$ a.u.) indicate the contribution of the donut shaped $|m|=1$ orbital at the yield minima, as seen at $\Delta t = T_{SO}$ in Fig. 2(b). The red area at small perpendicular momenta ($|p_y| < 0.5$ a.u.) indicates the dominance of ionization from the $m=0$ orbital at the yield maxima, as seen at $\Delta t = \frac{1}{2} T_{SO}$ in Fig. 2(b).”

Additionally, we have added the following statement on the calculation of normalized differences in the Methods section “Data analysis”:

“Having identified the electrons produced in the second ionization step, the electron density plots are obtained by calculating normalized differences of signal S , and reference R , photoelectron momentum distribution: $D = (S - R) / (S + R)$.”

Comment 8: *“In Fig. 2c, can you comment on why there is a discrepancy between theory and experimental results in the real space data at T_{SO} , particularly along the $y=0$ line. Secondly, at $1/2T_{SO}$, why is your measured distribution compressed along the x-axis?”*

These discrepancies are due to the distortion of the momentum maps caused by the mid-IR deflection field. Taking into account this distortion in the calculations (see comment #6 and #9) makes these effects also visible in the calculated distributions.

To pre-empt this question we are now spelling this relation out by adding a sentence to the manuscript on page 2:

“The stretch in momentum space corresponds to a contraction in the real space images.”.

Comment 9: *“At times $4/6$ and $5/6 T_{SO}$ in Fig. 2, do your calculations also reproduce the measured features in both momentum and real space $\times 2013$; please could you show these results, at least in the SOM. Can you also comment on the features present in the measured distributions of panels (b) and (c).”*

We included a new figure (Supplementary Figure 2) in the SI, section “Imaging model” that shows the calculated distributions for $4/6$ and $5/6 T_{SO}$. Note that these distributions look qualitatively identical to the ones at $\frac{1}{2} T_{SO}$ and T_{SO} but with lower contrast. This is consistent with the vacancy migrating from the $m=0$ to the $m=1$ states and back. We discuss this in the new SI section “Calculated electron density snapshots”. See also the response to Comment #6.

In the main text, we added the sentence

“At intermediate values $4/6 T_{SO}$ and $5/6 T_{SO}$, the images are qualitatively identical to the ones at $1/2 T_{SO}$ and T_{SO} , respectively, but exhibit a reduced contrast, see Supplementary Figure 2.”

Comment 10: *“In Fig. 3, you state ‘selected electrons’ but which selected electrons are you referring to?”*

The selected electrons are those that are removed from Ar^+ , i.e. fulfill the conditions defined in the methods (equation 1 and 2).

In the figure caption we replaced

“Momentum distributions in the p_y/p_z plane for selected electrons”

with

“Momentum distributions in the p_y/p_z plane for ionization from a coherent wave packet in Ar^+ .”

Comment 11: *“In Fig. 3(c), can you comment on the two outlier measured data points at p_z of 1-2.”*

This comment led us to revise what exactly is shown in Figure 3 (c) in relation to Figure 3(b). The former is integrated over the third dimension, while the latter had been integrated only over a small section of the third dimension. In the new Figure 3(c) that we provide, the data is integrated in the same manner as in Figure 3(b). The somewhat broader distribution agrees well with the calculations for the intensity of $1.0 \times 10^{15} \text{ Wcm}^{-2}$, rather than $0.6 \times 10^{15} \text{ Wcm}^{-2}$. Given the simplicity of the model the intensity mismatch is reasonable. Importantly, the conclusion from Figure 3(c), i.e. that the maxima observed at large longitudinal momenta can be attributed to the different ionization behaviour of the two fine structure states, remains intact.

In the text, we replaced

“The experimental results are selected for perpendicular momenta $\$p_x < 0.6 \backslash \mathrm{a.u.}\$ and compared to the results of a computational model”$

by

“The experimental results are selected for small perpendicular momenta (indicated by the dotted box in Fig. 3 (b) and compared to the results of a computational model”

Comment 12: *“In Fig. 4(d), can you please include error bars in your measured data.”*

We have added error bars to Fig. 4(d).

Comment 13: *“On page 4, you state ‘So far, we have shown that we can separate the first from the second photoelectron produced by sequential double ionization.’. Where have you shown this and explicitly discussed this? Is this related to your earlier statement of ‘This allows us to separate the*

primary and secondary photoelectrons spatially and thereby enables direct imaging of the valence-shell wave packet.’? Maybe I have missed this, but can you please clarify the statement on sequential double ionization?”

We agree that the statement is somewhat misleading. Clearly, we cannot separate the first and second photoelectrons in single-pulse sequential double ionization. Rather, we can identify those events in which the first ionization step occurs in the pump pulse and the second ionization step occurs in the probe pulse; and, for those events, we can separate the first from the second photoelectron. Thus, we have rephrased the sentence as follows.

We replaced

“So far, we have shown that we can separate the first from the second photoelectron produced by sequential double ionization”

with

“So far, we have shown that our pump-probe scheme allows us to identify double ionization events where the first and second ionization occurs in the pump and probe pulse, respectively. For these events we can separate the first from the second photoelectron, exploiting the deflection induced by the mid-IR streaking field [26].”

Comment 14: *“In the next sentence, you state that ‘the second photoelectron enables us to image the oscillating valence electron density’. However, nowhere in the main text above did you mention that it is the second electron photoelectron momentum distribution data in Fig. 2 and Fig. 3 that shows this. How can you distinguish the signal from the first and second electron in the Ar^{2+} momentum map signals? How can you be so sure and specific that the oscillating valence electron density signal is exclusively from the second photoelectron? It would be helpful for future readers to have this point clarified, at least with a brief explicit sentence.”*

The relevant physics are discussed in the fourth paragraph on page 1 (“Some of the simplest bound electron wave packets...”). The first ionization occurs from a stationary state in neutral Ar, whereas the second ionization occurs from the non-stationary state in Ar^+ . Thus, only the second electron carries the time-dependent information on the valence electron motion.

We rephrased the text in the fourth paragraph on page 1 to make it clearer that only the second electron carries the time-dependent information. We replaced

“As the spin-orbit wave packet evolves, the $3p^{-1}$ electron-hole density in the noble gas ion changes from a peanut shape into a donut shape and back, illustrated in Fig. 1. This oscillation in the spatial electron density leads to a modulation of the angle-dependent tunnel ionization probability [24]. The main obstacle to directly imaging the evolving electron-hole is the contamination of the signal with photoelectrons from the pump pulse [15]. In our approach we superimpose a weak, orthogonally polarized, carrier-envelope phase-stable, infrared streaking field \cite{Kubel2017} onto the probe pulse.”

with

“As the spin-orbit wave packet evolves, the $3p^{-1}$ electron-hole density in the noble gas ion oscillates between the $m=0$ state and the $|m|=1$ states (m being the magnetic quantum number). This oscillation leads to a time-dependent modulation in the angle-dependent

tunnel ionization probability of the ion [14]. Time-resolved measurements of the momentum distribution of photoelectrons, emitted from the ion, would allow for directly imaging the evolving electron-hole [24]. The main obstacle is the contamination of the signal with photoelectrons from the pump pulse [15]. In our approach we circumvent this obstacle by superimposing a weak, orthogonally polarized, carrier-envelope phase-stable, infrared streaking field [26] onto the probe pulse.”.

On page 4, we replaced

“Recording the transverse momentum distribution of the second photoelectron enables us to image the oscillating valence electron density in the argon cation.”

with

“Recording the transverse momentum distribution of the second photoelectron enables us to image the electron dynamics unfolding in the cation.”.

We hope that this clarifies the question raised here.

Comment 15: *“The first sentence in the second column of text on page 4: why ‘partially’? Why only partial reconstruction - can you comment please.”*

In the text, we discuss that reconstruction of the spatial distributions requires phase information that could be computationally reconstructed. Besides the missing phase information, the complete reconstruction of the spatial orbitals is precluded by the following limitations:

1. The present measurement is two-dimensional, as the valence electron density is projected onto the plane perpendicular to the laser polarization. Tomographic imaging could be applied to extend orbital imaging to three dimensions.
2. Tunnel ionization acts as a low-pass filter for electron momenta. This reduces the contrast at large momenta and limits the detectable momentum range to $< \sim 2$ a.u.. This sets a limit to the spatial resolution of approximately 0.5 a.u.

In the text, we replaced

“Thus, valence electron densities in real space could be partially reconstructed from time-resolved orbital imaging experiments, using an iterative algorithm. Such endeavors require very good statistics in the experimental data, and will greatly benefit from the development of few-cycle laser sources approaching the MHz range [31].”

with

“Thus, valence electron densities in real space could be reconstructed from time-resolved orbital imaging experiments, using appropriate computational techniques. The spatial resolution of the reconstructed real-space distribution is given by the maximum perpendicular photoelectron momentum, which is limited by the transversal filter function and the signal to noise ratio. Time-resolved orbital imaging will greatly benefit from the development of intense few-cycle laser sources approaching the MHz range [31].”.

Comment 16: *“You mention coulomb explosion imaging (CEI) and laser-induced electron diffraction (LIED) at the end of this paragraph. Can you please include some referenced work, such as*

CEI1 - ultrafast nuclear dynamics: K. Amini et al., Structural dynamics 5, 014301 (2018).

CEI2 - geometric isomer configurations: M. Burt et al., J. Chem. Phys. 148, 091102 (2018).

LIED1 - imaging deprotonation reaction: Wolter2016 reference used in this manuscript.

LIED2 - intra-pulse nuclear-electronic dynamics: K. Amini et al., arxiv:1805.06793.”

We included the suggested references in the final paragraph of the main text.

Comment 17: *“In your methods, can you please specify the pulse energy and focal spot size ($1/e^2$ width) of your three laser pulses.”*

We added this information in the second paragraph of the Experimental setup section in the Methods.

Comment 18: *“Throughout the manuscript, you use the word ‘via’ quite regularly, particularly in the top-right paragraph on page one. Please avoid so many repetitions of this word in such a short space of text. Also, please make ‘via’ italicized.”*

We have replaced 5 out of 8 occurrences of ‘via’ with alternative words.

Comment 19: *“Please make all roman variables in the main text (such as momentum, p) italicized.”*

We followed the referee’s advice and italicized all variables in the main text.

Further changes

1. As required by Nature Communications, we have removed citations from the abstract.
2. We have added sections (Introduction, Results, Discussion)
3. At the end of the fourth paragraph of the introduction, we have added a brief summary of results:

“We reconstruct features of the spatial valence electron density from the recorded momentum space images and discuss its full reconstruction. Moreover, we show how the varying orbital shape affects the sub-cycle ionization behavior of the ion.”

4. We have included two Supplementary movies showing experimental results.

We have added a reference to Movie 1 in the second paragraph of the Results:

“In Figure 2 (b) and Supplementary Movie 1, we present a time series of measured electron density plots for the wave packet in Ar⁺.”

We have added a reference to Movie 2 in the second paragraph of the Discussion:

“Figure 4(a) and Supplementary Movie 2 show the normalized differences...”

REVIEWERS' COMMENTS:

Reviewer #1 (Remarks to the Author):

The authors addressed my only concern very adequately. As I had no other misgivings, I can now recommend publication of this manuscript in Nature Communications.

Reviewer #2 (Remarks to the Author):

Thank you for the revised manuscript and replies to the referee comments. I fully support the publication of this revised manuscript, with an excellent manuscript and detailed response to referee comments.

I have two very minor points to make which may be typos in the text to help the authors:

- Comment 6: In your SI under section “Calculated electron density snapshots”, you mention that “the Ar⁺ valence orbital are obtained as described in section Imaging model above”, but you mean they are shown in the section below and not above.
- Comment 7: do you mean to have the dollar signs in “ $|m|=1$ ”? A sentence earlier or so, you have $|m|=1$ without dollar signs.

Dr Kasra Amini

Reply to Referee #2:

Comment 6: "In your SI under section "Calculated electron density snapshots", you mention that "the Ar+ valence orbital are obtained as described in section Imaging model above", but you mean they are shown in the section below and not above."

We have corrected the sentence as suggested.

Comment 7: "do you mean to have the dollar signs in "\$|m|=1\$"? A sentence earlier or so, you have |m|=1 without dollar signs."

We mean to typeset all mathematical expressions, such as $|m|=1$ in the math environment and have made sure that we do so.